# Soluble Factors and Receptors Involved in Skin Innate Immunity—What Do We Know So Far?

**DOI:** 10.3390/biomedicines9121795

**Published:** 2021-11-29

**Authors:** Lucian G. Scurtu, Olga Simionescu

**Affiliations:** Department of Dermatology, “Carol Davila” University of Medicine and Pharmacy, Colentina Hospital, 19-21 Stefan cel Mare Road, 020125 Bucharest, Romania; lucian.scurtu@drd.umfcd.ro

**Keywords:** innate immunity, pattern recognition receptors, cytokines, inflammasomes, complement, antimicrobial peptides, imiquimod, biologics

## Abstract

The pattern recognition receptors, complement system, inflammasomes, antimicrobial peptides, and cytokines are innate immunity soluble factors. They sense, either directly or indirectly, the potential threats and produce inflammation and cellular death. High interest in their modulation has emerged lately, acknowledging they are involved in many cutaneous inflammatory, infectious, and neoplastic disorders. We extensively reviewed the implication of soluble factors in skin innate immunity. Furthermore, we showed which molecules target these factors, how these molecules work, and how they have been used in dermatological practice. Cytokine inhibitors have paved the way to a new era in treating moderate to severe psoriasis and atopic dermatitis.

## 1. Introduction

The skin is the largest organ of the human body and is subsequently fully equipped as the main barrier against aggressive environmental factors. The epidermis, the dermis, and the hypodermis are the three layers that define this complex organ. As a physical barrier, the epidermis has 90–95% keratinocytes and a few melanocytes, Langerhans cells, Merkel cells, and resident memory CD8+ T cells. The dermis, on the other hand, consists of a heterogeneous population of cells and provides the role of nutrition, defense, and support for the upper layer. Dermal innate lymphoid cells secrete molecules that serve as innate immunity soluble factors [1,2]. The lack of corneocytes in the skin appendages, such as the hair follicles, brings together the microbiota and the keratinocytes and promotes a unique immunological environment [3]. Both innate and adaptive immune responses can be triggered within the skin. Pattern recognition receptors (PRRs), complement system, inflammasomes, antimicrobial peptides, and cytokines are soluble factors involved in innate immunity [1]. PRRs activation results in a signal transduction cascade that promotes activation of transcription factors which may modify cytokines expression. They have a significant role in frequent dermatoses such as psoriasis and atopic dermatitis [3,4,5]. Specific defects in the inflammasome complex may lead to cryopyrin-associated periodic syndrome. Novel interleukin inhibitors have changed the therapeutic approach for this disorder [4]. During their differentiation process, keratinocytes produce antimicrobial peptides. Both antimicrobial peptides and PRRs are potent barriers against pathogens [3,4,6]. Complement system activation and inflammatory cytokine production represent essential steps in skin innate immunity activation and may trigger the adaptive immune system by a complex, self-amplification mechanism [7]. The literature lacks a comprehensive approach regarding soluble factors and receptors involved in skin innate immunity and the novel drugs that target these factors. The purpose of this review is to provide a more thorough knowledge of these biomolecules and their role in the management of patients with skin diseases.

## 2. Pattern Recognition Receptors

Fungi, protozoa, viruses, and bacteria share common pathogen-associated molecular patterns (PAMPs), which are identified by the innate immune system via highly specialized receptors, the so-called “pattern recognition receptors” (PRRs). These receptors are exposed both on immune and non-immune cells and serve as a first-line defense against PAMPs. Four major families of pattern recognition receptors have been described yet: Toll-like receptors (TLRs), RIG-I-like receptors (RIGs), NOD-like receptors (NLRs), and the C-type lectin receptors. The innate immune system uses PRRs to distinguish between self entities and microorganisms. Autoimmune diseases may be linked to PRRs recognition of self molecules, such as nucleic acids [4,5,6]. For example, the aberrant activation of the innate immune response in cutaneous lupus erythematosus is triggered by the cellular debris, rich in endogenous immunostimulatory nucleic acids which bind to PRRs and induce interferon responses and inflammasome activation [7].

TLRs are a family of glycoproteins usually expressed by granulocytes, dendritic cells, macrophages, and monocytes. Keratinocytes may express TLRs as well. TLRs are located on the cell surface, in the cellular plasma membrane, and in intracellular compartments. They contain ectodomains that mediate PAMPs recognition and transmembrane domains and endo-domains necessary for downstream signal transduction [6]. Apoptosis, shaping the development of the adaptive immune response, and triggering the release of antimicrobial peptides are important TLRs activation consequences [8].

While TLRs recognize endosomal nucleic acids, the retinoic acid-inducible gene (RIG)-I-like receptors (RIGs) detect cytosolic viral RNA and induce type I interferon production. Twenty-two intracellular NOD-like receptors (NLRs) classify into four subfamilies based on their N-terminal domain. NOD1 and NOD2 receptors recognize peptidoglycans from the bacterial wall and finally promote pro-inflammatory mediators’ expression, while NLRP3 activation leads to inflammasome assembly.

C-type lectin receptors are glycosylated type-II transmembrane PRRs that promote anti-fungal effects via their carbohydrate-recognition domains (CRDs). Representative members of this family include dendritic cell-associated C-type lectin 1 (Dectin 1), Dectin 2, macrophage C-type lectin (MCL), and macrophage inducible C-type lectin (Mincle). Mincle is capable to respond to *C. albicans* and Malassezia species (Figure 1) [5,6,7].

## 3. Complement

Osmotic cell lysis is the final consequence of the complement pathway activation (Figure 2). As described by Paul Erlich, in the year 1899, the complement system encompasses more than 20 serum proteins that act as proteases [8,9]. Human keratinocytes and skin fibroblasts can produce many complement proteins and act as a strong host defense barrier. Human keratinocytes produce C3, C5, C7–C9, factor B, complement receptors, the complement inhibitors factor H, and factor H-like protein-1 (FHL-1) [10,11].

## 4. Inflammasomes

Inflammasomes are high-molecular-weight proteins that induce cell programmed death called pyroptosis (Figure 3). Inflammasomes are usually found in the cytoplasm of activated immune cells and work as a sensor for monitoring PAMPs and DAMPs (danger-associated molecular patterns). They consist of a central sensor protein (that is usually a Nod-like receptor or a hematopoietic IFN-inducible antigen (HIN) domain protein), an adaptor protein ASC (apoptosis-associated speck-like protein containing a caspase recruitment domain), and an effector protein, which is caspase-1. The most common central sensor Nod-like receptors are: NLRP1, NLRP3, NLRC4, and NLRP6, and the most common HINs are: AIM2 and IFN-γ-inducible protein-16. Other central sensor proteins that can form inflammasomes are pyrin, TLRs, and retinoic acid-inducible gene (RIG) receptors [12,13,14].

The NLRP-3 inflammasome (also known as cryopyrin) responds to viral, bacterial, fungal, and protozoan pathogens, extracellular ATP, pore-forming toxins, uric acid crystals, and silica. This large set of stimuli suggests that NLRP-3 cannot be a direct sensor for all these agents. However, some stimuli such as potassium efflux, lysosomal destabilization, production of reactive oxygen species, and the formation of membrane pores have been described [15]. Among these stimuli, potassium efflux is a common step essential for NLRP-3 activation, but the exact mechanism remains unknown. NEK-7 is a mammalian kinase that was shown to display a role downstream of potassium efflux to facilitate NLRP-3 formation and activation [16].

Several hereditary periodic fever syndromes such as familial cold autoinflammatory syndrome (FCAS), Muckle–Wells syndrome, and neonatal-onset multisystem inflammatory disease (NOMID) are cryopyrin-associated periodic syndromes (CAPS). Autosomal dominant mutations in the NLRP-3 encoding gene lead to caspase-1 activation and abnormal formation of IL-1-β, clinically presenting with fever, urticaria-like lesions, and joint inflammation. It was proved that selective and prolonged IL-1-β blockade using a fully human monoclonal antibody (e.g., canakinumab, anakinra) provides disease control in children and adults with CAPS [17,18]. DNA release in the cytosol following rupture of the nuclear envelope promotes AIM2 inflammasome activation, suggesting its role in monitoring nuclear integrity. AIM2 activation displays a particular role in psoriasis [14,19].

## 5. Antimicrobial Peptides

Antimicrobial peptides (AMPs) are an evolutionarily preserved family of peptides found both in vertebrates and invertebrates that consist of 20–60 amino acids in length [20]. AMPs display the capacity to announce infection or injury; therefore, the term “alarmins” is common for some AMPs. AMPs were initially described in the wound repair process, and further research has proved their bactericidal effect and capacity to alert the host and activate the immune system [21].

This bactericidal mechanism is due to the cationic structure of most of the AMPs that attach to the anionic bacterial cell surface, as described for group A *Streptococcus* and other bacteria [22]. Defensins, cathelicidins, dermcidins, and other molecules, such as RNase 7, psoriasin, and lactoferrin, are among AMPs produced in human skin [20]. Although more than 20 different proteins were proven to provide antimicrobial activity, cathelicidins and β-defensins are the most precisely described AMPs [23].

Both cathelicidins and β-defensins are negligible in normal skin. Per contra, an increased expression in the superficial epidermis was shown in psoriasis. A lower expression was found in eczematous lesions from atopic dermatitis patients, which may be significant in terms of their susceptibility to developing skin infections with gram-positive germs [24]. AMPs have demonstrated an important role in wound healing and re-epithelization, by increasing angiogenesis, cell proliferation, and preventing pathogen multiplication in the injured tegument [25]. Furthermore, cathelicidins inhibition was linked with impaired re-epithelialization in wounded organ-cultured human skin [26].

## 6. Cytokines

Cytokines represent a family of small proteins (<40 kDa) released by most of the cells, mainly macrophages and helper T cells, that display a specific effect on the immune response. They can act on distant cells, nearby cells (a paracrine mechanism), or the same cells that secrete them (an autocrine mechanism) [27,28]. Both pro-inflammatory and anti-inflammatory cytokines are secreted during the immune response, however, their effects depend on the targeted cells [29,30,31].

Keratinocytes also produce many cytokines, such as interleukin-1 (IL-1), interleukin-6 (IL-6), interleukin-7 (IL-7), interleukin-8 (IL-8), interleukin-10 (IL-10), interleukin-12 (IL-12), interleukin-15 (IL-15), interleukin-18 (IL-18), interleukin-20 (IL-20), and tumor necrosis factor-alpha (TNF-α) [32]. It was proven that various environmental molecules such as sodium lauryl sulfate, phenol, and croton oil induce IL-8 and other proinflammatory cytokines expression, production, and release in human keratinocyte cultures, suggesting their importance in skin inflammation [33]. HPV infection, especially high-risk strains, was linked with increased expression of IL-6 in infected keratinocytes promoting local tumorigenesis [34].

Melanocytes are neuroimmune cells that express type I IFN as an early innate response to viruses. They respond against pathogens using PRRs and can initiate cell death accompanied by IFN-b, TNF-a, IL-6, and IL-8 expression, as a response to some viral analogs [35]. Nevi and thin primary melanomas show little expression of TNF-α, transforming growth factor-β (TGF-β), and IL-8, while a marked-up regulation of cytokines and growth factors can be noticed in thick primary melanomas and metastatic melanomas [36]. The melanoma inhibitor of apoptosis protein (ML-IAP) is an important regulator of apoptosis that inhibits caspase activation and TNF signaling, frequently overexpressed in melanoma cells, but with low expression in normal melanocytes. Expression of ML-IAP has been proved to enhance apoptosis resistance in melanoma by binding caspases 3, 7 and 9, and to provide melanoma cells a higher endurance during tumor progression [37]. Although melanoma cells possess this apoptosis-escape mechanism, the roles of non-apoptotic cell death signaling pathways, such as autophagy-dependent cell death, necroptosis (caspase-independent cell death, similar to necrosis), ferroptosis (iron-dependent lipid-peroxides induced cell death), pyroptosis (via inflammasomes), and parthanatos (mitochondrial accumulation of poly(ADP-ribosyl)-ated proteins’ induced cell death) are of major importance as well [38].

Many cytokines produced by other resident skin cells such as macrophages, NK cells, dendritic cells (Figure 4), or endothelial cells, act at numerous levels to control the cornification process, intercellular adhesion, or the expression of some genes that encode important enzymes. Skin barrier regulation is dictated by the interaction between ligands (cytokines) and cytokine receptor complexes, activating several signaling pathways such as Janus kinase/signal transducer and activator of transcription (JAK/STAT), phosphoinositide 3-kinase (PI3K/AKT), mitogen-activated protein kinases (MAPK) and nuclear factor kappa-light-chain-enhancer of activated B cells (NFκB) [31,39].

## 7. How Does Imiquimod Target PRRs?

In addition to their role in preventing infection, TLRs possess both tumorigenic and anti-neoplasia effects. TLRs expressed on the cancerous cells’ membrane promote invasion and metastasis in some cancers, while they inhibit neoplasia progression in others [40].

Initially known as S-26308 or R-837, Imiquimod (1-(2-methyl propyl)-1H-imidazo-[4,5-c]-quinoline-4 amine) is an imidazoquinoline amine analog to guanosine (Figure 5), approved for the treatment of actinic keratosis (AK), superficial basal cell carcinoma (BCC) and anogenital warts (*Condyloma acuminatum*), but is widely used in some other skin disorders as well. Imiquimod binds to the TLR-7 localized on the surface of Langerhans cells, dendritic cells, and monocytes and induces TNF-α and IFN-α secretion, proving its antiviral and antitumor effect [41]. Huang et al. previously showed that Imiquimod could induce immunogenic cell death (ICD) in vivo and in vitro. Imiquimod-induced ICD consists of reactive oxygen species (ROS) production in cancer cells. ROS trigger endoplasmic reticulum stress and subsequently promote surface exposure of calreticulin, ATP-dependent autophagy, and postapoptotic release of high mobility group box-1 protein (HMGB1). HMGB1 eventually promotes maturation of antigen-presenting cells and tumor antigen presentation [42].

### 7.1. Imiquimod in Basal Cell Carcinoma (BCC)

A 2019 meta-analysis on 4256 patients compared the safety and efficacy of imiquimod with other treatments in patients with BCC (including vehicle, excisional surgery, cryosurgery, 5-fluorouracil, methylamino levulinate photodynamic therapy). Imiquimod associated significant higher histological clearance rate (RR = 9.28, 95% CI: 5.56, 15.49; *p* < 0.001) and complete response rate (RR = 3.15, 95% CI: 1.55, 6.38; *p* = 0.001). Imiquimod was slightly inferior when comparing the success rates (RR = 0.93, 95% CI: 0.90, 0.96; *p* < 0.001). The authors suggest that imiquimod should be used as the first choice of treatment for BCC [43]. A multi-center, non-inferiority randomized controlled trial (RCT) with 501 participants with nodular and superficial basal cell carcinoma who were divided into two groups (imiquimod group, 254 participants, and surgical excision group, with 247 participants) concluded that there is no obvious difference between groups regarding the cosmetic outcome, but treatment successfulness for imiquimod was inferior to surgery (88% in imiquimod group, comparative to 98% in surgery group) [44]. According to the European consensus published in 2019, Imiquimod is particularly useful in low-risk single or multiple superficial BCC, with limited evidence on nodular BCC, but with a better response comparative to methyl-aminolevulinate photodynamic therapy for superficial BCC [45]. 

### 7.2. Imiquimod in Actinic Keratosis

Ogawa et al. used an antibody specific to plasmacytoid dendritic cells to investigate whether imiquimod stimulates these cells’ recruitment to AK lesions. Histological staining demonstrated that plasmacytoid dendritic cells were significantly more abundant after the topical treatment and their number inversely correlated with healing duration, indicating that these cells are critically involved in the regression of AK during imiquimod treatment [46].

In a randomized trial, imiquimod cream 5% effectiveness in AK was compared with other three topical treatments (ingenol mebutate gel 0.015%, fluorouracil cream 5%, methyl aminolevulinate photodynamic therapy). The study concluded that 5% fluorouracil was significantly more effective than imiquimod, MAL-PDT, or ingenol mebutate at 12 months for multiple AKs [47]. In a meta-analysis of five randomized double-blinded trials treating 1293 patients with AK (with almost 90% men) complete clearance occurred in 50% of patients treated with imiquimod and most adverse events were local, most frequently erythema, scabbing, and flaking, but serious adverse events were not significantly different between imiquimod or vehicle control, with a relative risk of 1.2 (0.7–2.0) [48].

A 2015 article suggested that imiquimod cream 3.75% is the only effective treatment across a large sun-exposed field (full face or balding scalp), for both clinical and subclinical AK (unmasked by imiquimod during treatment). The median percentage decline in lesions was 92%, and the efficient clearance of both clinical and subclinical lesions led to unremitting lesion clearance for at least 12 months. The authors concluded that imiquimod can reveal patients’ entire burden of disease and SCC risk [49].

### 7.3. Imiquimod in Condyloma Acuminatum

A 2020 meta-analysis included all the RCTs that evaluated topical treatments for external genital warts and revealed podophyllotoxin 0.5% solution is significantly more effective compared to imiquimod 5% cream. However, it was associated with a higher overall adverse event rate. None of the above treatments were significantly different from each other in terms of relapse [50].

In genital (penile) warts, imiquimod clearance rates are similar to those of podophyllotoxin, if applied three times per week until total clearance or for a maximum of 16 weeks for imiquimod cream 5%, respectively every night for up to 8 weeks for imiquimod cream 3.75%. The treatment surface should be washed six to ten hours after application to minimize local adverse events [51]. In an RCT comprising 228 patients, an eight-week combination therapy of 400mg oral zinc sulfate with topical imiquimod for vulvar warts significantly decreased relapse after six months [52].

### 7.4. Imiquimod Off-Label Use

Off-label use of imiquimod is not rare in our daily practice. Imiquimod possesses an immunomodulatory effect in neoplastic, infectious, and inflammatory skin disorders [53].

#### 7.4.1. Bowen Disease

Several clinical studies and case reports investigated imiquimod efficiency in Bowen’s disease. A retrospective study of 49 patients (96% males, 96% Caucasians, various body locations) concluded that 86% had a complete clinical response with topical imiquimod [54]. In an open-label study, 16 patients with a single biopsy-proven lesion that was 1 cm or larger in diameter (up to 5.4 cm) were assigned to once-daily self-application of imiquimod 5% cream for 16 weeks. Histological clearance was obtained in their 6-week posttreatment biopsy specimens in 93% of the lesions [55]. However, more extensive studies are necessary to compare imiquimod with other nonsurgical therapies in Bowen disease.

#### 7.4.2. Bowenoid Papulosis

Resembling *Condyloma acuminatum*, but with the histopathological appearance of Bowen’s disease, Bowenoid papulosis of the genital area can display a good response to topical imiquimod 5% [56], but the results are scarce, in a few case reports. An almost complete regression to topical imiquimod in monotherapy was obtained in HIV patients [57]. Favorable results were obtained in HIV-positive Bowenoid papulosis patients when combining oral acitretin with topical 5% imiquimod [58].

#### 7.4.3. Erythroplasia of Queyrat

With the cost of moderate burning sensation, imiquimod 5% can be an alternative for surgical and invasive procedures, which may provide poor cosmetic and functional outcomes since this type of in situ carcinoma is usually located on the glans of the penis [59]. In negative polymerase chain reaction (PCR) testing for HPV, imiquimod efficiency is most likely explained by cytokines’ activation, with antitumor effect, as shown in BCC [60].

#### 7.4.4. Keratoacanthoma (KA)

Complete surgical excision is the treatment of choice, but poor cosmetic outcomes may arise. Thus, conservative treatments using podophyllin, 5-fluorouracil, imiquimod, or even intralesional injection of methotrexate or corticosteroids may be beneficial. Topical imiquimod 5% was successfully used in some solitary facial KA [61], and low-dose acitretin association was documented to promote complete resolution in multiple KAs patients [62]. Contrariwise, a retrospective study showed that conservative treatments led to significantly higher recurrence rates when compared to surgical treatment [63].

#### 7.4.5. Squamous Cell Carcinoma (SCC)

The available studies do not currently endorse topical treatments in SCC; therefore, surgical excision is the preferred treatment [64]. When applied over large surfaces, inflammation may occur, decreasing topical imiquimod usefulness in field cancerization. 5-fluorouracil is preferred instead [65].

#### 7.4.6. In Situ Melanoma

Topical imiquimod 5% can be used as a second-line treatment in lentigo maligna, particularly in large tumors, which are located on the face, ears, or scalp, especially in poor surgical candidates and old patients. Adjuvant imiquimod after complete surgical excision showed improved outcomes in terms of local recurrence [66]. Other skin malignancies that may respond to topical imiquimod include extramammary Paget disease (alternative treatment to surgery) [67], the classical form of Kaposi disease (safe and suitable for home application) [68], folliculotropic mycosis fungoides [69], or melanoma skin metastases (palliative therapy, favors metastases resolution) [70].

#### 7.4.7. Molluscum Contagiosum

On the one hand, a recent case-control study on 48 pediatric patients recommended topical imiquimod as a first-line treatment in children with disseminated lesions [71]. On the other hand, some scientists found the use of imiquimod in children with *molluscum contagiosum* not only inefficient but also hazardous, considering high systemic absorption, hematological abnormalities [72], and even pemphigus-like eruption as side-effects [73].

#### 7.4.8. Herpes Simplex Virus (HSV)

Acyclovir is a drug extensively used to treat HSV infections. Some recent studies have even investigated vaginal acyclovir delivery methods [74]. In a few case reports, immunocompromised patients with acyclovir-resistant anogenital herpes simplex were reported to respond to topical administration of imiquimod, thus avoiding the intravenous administration of toxic antiviral drugs, such as cidofovir or foscarnet [75]. Imiquimod interacts with the adenosine receptor A1, activating the protein kinase A pathway and upregulating cystatin A synthesis in infected nonimmune cells, proving an IFN-independent antiviral mechanism [76]. Herpes simplex virus (HSV) can manipulate TLR and evade the innate immune response. Recent studies showed that HSV-mediated TLR signaling enables virus replication and suppresses interferon production via NF-κB signaling [77]. Another double-stranded DNA virus, varicella-zoster virus, can induce proinflammatory cytokines upon activation of TLR-2, while TLR-2 abnormalities were associated with an increased number of local recurrences in patients with genital HSV infection [78].

#### 7.4.9. Porokeratosis

Several case reports proved that porokeratosis of Mibelli has a favorable response to topical imiquimod 5%, both in children and elders, including some severe cases, nonresponsive to prior cryotherapy or 5-fluorouracil [79,80]. It is also beneficial in some patients with disseminated or linear porokeratosis. Some authors suggest that imiquimod may modulate p53 tumor suppression protein and have reported overexpressed or mutant p53 protein in porokeratosis lesions; therefore, a p53-dependent mechanism may explain imiquimod efficiency in porokeratosis [81,82].

#### 7.4.10. Morphea

Imiquimod’s effectiveness in morphea is attributed to its capacity to induce local interferon synthesis, which inhibits collagen production by human fibroblasts, as shown in other sclerodermoid disorders. A 2011 prospective, open-label, double-baseline study showed that topical imiquimod 5% is effective and safe in children with plaque morphea. A decrease in the thickness of the plaques was quantified using ultrasound [83]. Contrarily, a 2015 study that enrolled 25 adult patients with plaque morphea showed no ultrasonographic differences in thicknesses between the treatment and vehicle groups, but induration demonstrated a favorable response [84].

#### 7.4.11. Discoid Lupus Erythematosus (DLE)

Following imiquimod 5% treatment, patients with generalized DLE showed remission, after 20, respectively, 24 applications, in two case reports [85,86]. Nevertheless, imiquimod-induced cutaneous lupus erythematosus has been reported in the literature [87].

## 8. Targeting Other Soluble Factors

Complement pathways are associated with several skin disorders, such as bullous pemphigoid, psoriasis, urticaria, urticarial vasculitis, hidradenitis suppurativa, vasculitis, and lupus erythematosus. Consequently, interest in producing complement-specific molecules has been increasing [88]. Intravenous immunoglobulin (IVIG) binds C1q, C3, and C4, preventing C1 attack and C3 and C4 in situ deposition, respectively. IVIG also downregulates C3 convertase activity, therefore it is successfully used in some small-medium-vessel vasculitides [89]. Eculizumab and its long-lasting form, ravulizumab, are monoclonal antibodies that bind to C5 and inhibit its cleavage by C5a, both approved for paroxysmal nocturnal hemoglobinuria [90]. IFX-1, an anti-C5a inhibitor, was evaluated in a trial involving 12 participants with hidradenitis suppurativa (NCT03001622); however, no results are currently available [91].

Canakinumab is an interleukin-1 monoclonal human antibody that can successfully control and prevent flares in patients with colchicine-resistant familial Mediterranean fever, mevalonate kinase deficiency, and tumor necrosis factor receptor-associated periodic syndrome, as proved in a 2018 study [92]. Canakinumab is additionally effective in Schnitzler syndrome, with the cost of infection as a frequently encountered adverse event [93], and can also represent a therapeutic option for steroid-resistant pyoderma gangrenosum [94]. Some cases of pyoderma gangrenosum remain refractory to this treatment [95]. Anakinra is an IL-1 inhibitor that demonstrated a decreased disease activity of moderate to severe hidradenitis suppurativa in a five-patient open-label study [96].

Other interleukin inhibitors have been approved for atopic dermatitis and psoriasis (Figure 6). Dupilumab is an IL-4/IL-13 inhibitor approved in moderate-to-severe atopic dermatitis [97]. Risankizumab, guselkumab, tildrakizumab (p19 subunit of IL-23 inhibitors), ustekinumab (IL-12/23 inhibitor), ixekizumab, brodalumab, and secukinumab (IL-17 inhibitors) are the modern biologics approved in moderate-to-severe plaque psoriasis. TNF-α inhibitors, such as infliximab, adalimumab, certolizumab, and etanercept, may similarly represent a good alternative in patients with moderate-to-severe psoriasis [98,99,100,101]. Most biologics used in psoriasis show no difference regarding short-term efficacy and tolerability [102].

## 9. Conclusions

As far as we are aware, this is the first extensive review regarding the soluble factors involved in skin-innate immunity and the drugs that target these factors. Imiquimod, a TLR-7 ligand, is currently approved for BCC, AK, and *Condyloma acuminatum*, but there is an emerging interest in using imiquimod for other skin cancers, infections, and inflammatory skin conditions. Nevertheless, future prospective studies are warranted to provide clear guidelines regarding the off-label uses of Imiquimod. Newly discovered IL-1 inhibitors, canakinumab and anakinra, exhibit a consistent inhibition of the abnormal inflammasome assembly, providing optimistic results in managing patients with CAPS and Schnitzler syndrome. Cytokine inhibitors have opened the door to a new era in treating moderate to severe psoriasis and atopic dermatitis. Given their role in a variety of dermatological diseases, further research is warranted. Future investigations concerning soluble mediators and receptors of the cutaneous innate immunity may thoroughly emphasize their action mechanisms and contribute to new treatment protocols.

## Figures and Tables

**Figure 1 biomedicines-09-01795-f001:**
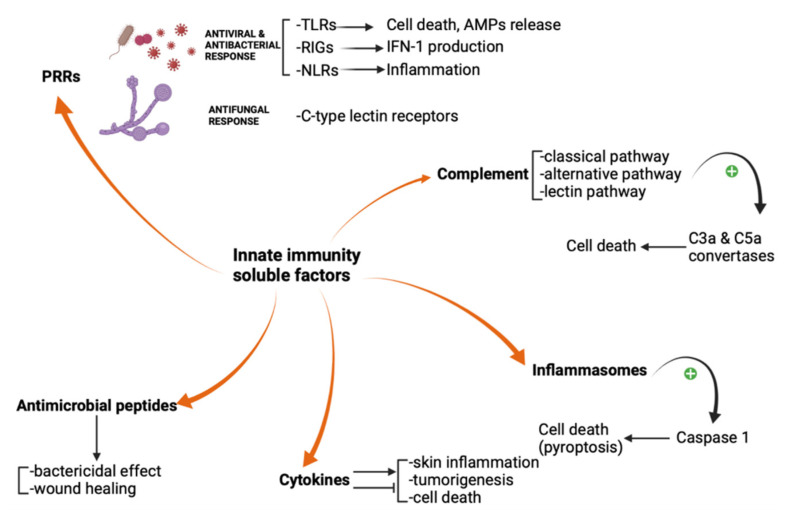
The main pathways of the soluble factors involved in skin innate immunity. PRRs, antimicrobial peptides, cytokines, inflammasomes, and complement system act as a first defense line against pathogens. PRR = pattern recognition receptor; TLR = Toll-like receptor; RIG = RIG-I-like receptor; NLR = NOD-like receptor.

**Figure 2 biomedicines-09-01795-f002:**
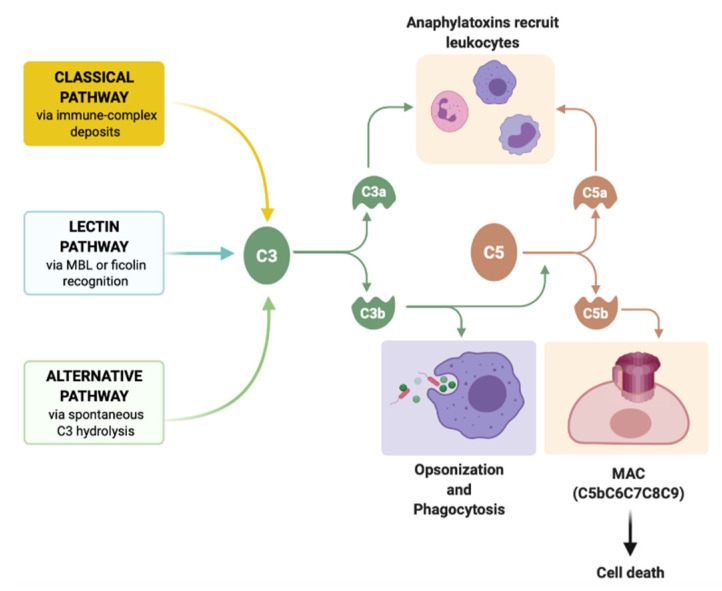
Complement activation cascade. Three major different complement activation routes were described so far: the classical pathway, the lectin pathway, and the alternative pathway. All these routes converge at the activation of C3a and C5a, which lead to the formation of the membrane attack complex (MAC, C5bC6C7C8C9) and consecutive cell lysis (apoptosis). The lectin pathway activates by mannose-binding lectin (MBL) or ficolin recognition of some sugar molecules exposed on the surface of microorganisms. The classical pathway and the alternative pathway act antibody-dependent, respectively antibody-independent, but similarly with the lectin pathway, they activate the C3a and the C5a convertases. Anaphylatoxins (C3a and C5a) act as leukocytes recruiters within the skin, while C3b deposition induces opsonization, phagocytosis and clearance of apoptotic debris, and immune complex deposits.

**Figure 3 biomedicines-09-01795-f003:**
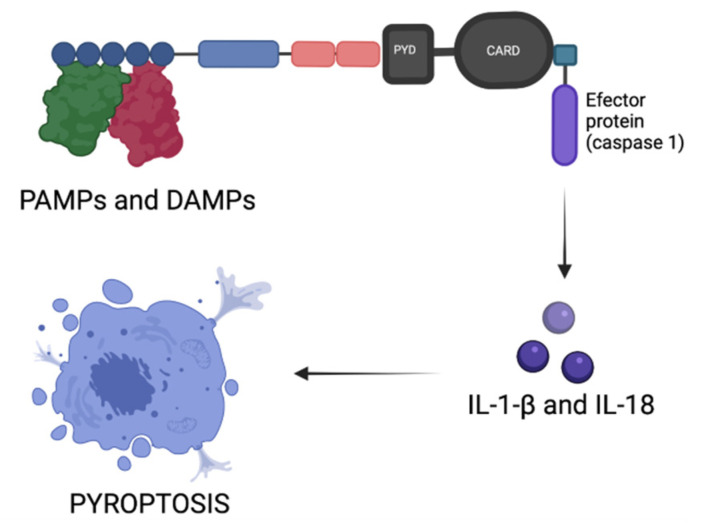
Inflammasome assembly and pyroptosis. In response to PAMPs and DAMPs, the central sensor recruits the adaptor protein ASC, which contains two death-fold domains: a caspase recruitment domain (CARD) and a pyrin domain (PYD), that bind to the inflammasome sensor. Consecutively, ASC induces the formation of the catalytically active effector protein caspase 1, which initiates IL-1-β and IL-18 release and pyroptosis of the infected cell. PAMPs = pathogen-associated molecular patterns; DAMPs = danger-associated molecular patterns.

**Figure 4 biomedicines-09-01795-f004:**
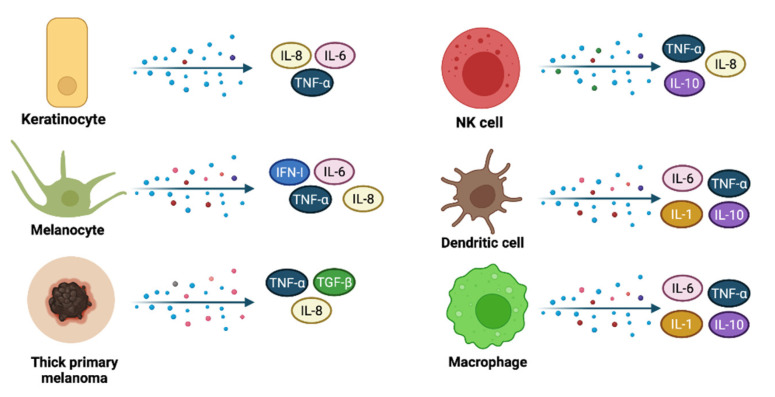
Cytokines release during skin immune response and in thick primary melanomas. Keratinocytes, melanocytes, and other innate immunity cells, such as NK cells, dendritic cells and macrophages secrete cytokines that may serve both as pro-inflammatory and anti-inflammatory molecules or may trigger carcinogenesis (IL-6). Thick primary melanomas secrete IL-8, TNF-α, and TGF-β, suggesting a high synthesis rate within neoplastic cells.

**Figure 5 biomedicines-09-01795-f005:**
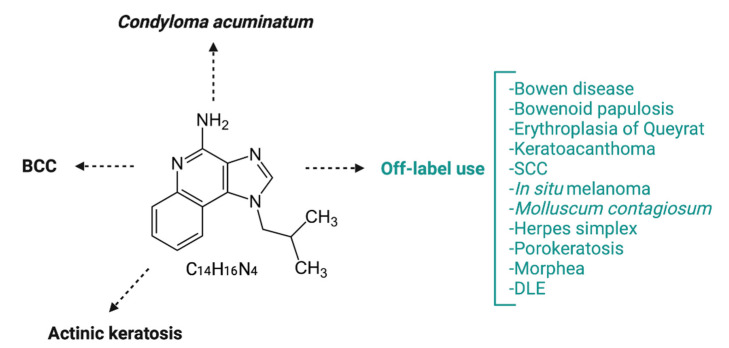
Imiquimod chemical structure (central) and approved and off-label use of Imiquimod in skin diseases. Imiquimod is an amine analog to guanosine, currently approved for topical treatment of BCC, KA, and anogenital warts. Nowadays, in dermatological practice, imiquimod is an off-label drug used to treat many skin diseases, such as inflammatory skin disorders, neoplastic diseases, or skin infections. SCC = squamous cell carcinoma; DLE = discoid lupus erythematosus.

**Figure 6 biomedicines-09-01795-f006:**
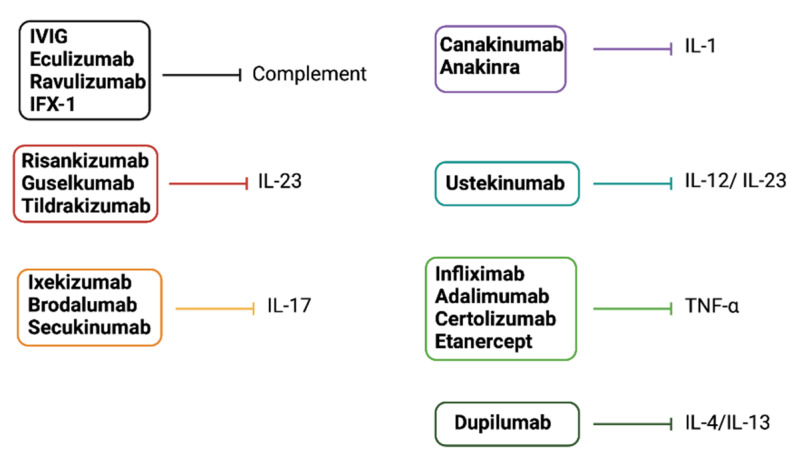
Drugs that target innate immunity soluble factors (others than PRRs). Complement modulators and biologics target innate immunity soluble factors. Dupilumab inhibits IL-4/IL-13 and is approved for use in moderate-to-severe atopic dermatitis. IL-23, IL-17, IL-12/IL-23 inhibitors, and TNF-α inhibitors are approved for use in moderate-to-severe plaque psoriasis. Canakinumab and anakinra inhibit IL-1. Eculizumab, ravulizumab, IFX-1, and IVIG inhibit complement pathways.

## Data Availability

Not applicable.

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
