# Peer review of "Soluble Factors and Receptors Involved in Skin Innate Immunity—What Do We Know So Far?"

_biomedicines, 2021, doi:10.3390/biomedicines9121795_

Round 1

Reviewer 1 Report

The manuscript is written in an acceptable way. I recommended some comments to improve the quality of the manuscript for the readers of the journal.

  • The introduction section is too short and needs to rewrite in a new form.
  • Section 6. Please bring the full form of the following:

IL-1, IL-6, IL-7, IL-8, IL-10, IL12, IL-15, IL-18, IL-20

  • Section 7.4.8

One of the main drugs to treat HSV is known as Acyclovir which has been used as a drug in vaginal drug delivery studies. I recommend to the authors to use the following reference in this manuscript also.

Sabbagh, F., & Muhamad, I. I. (2017). Acrylamide-based hydrogel drug delivery systems: release of acyclovir from MgO nanocomposite hydrogel. Journal of the Taiwan Institute of Chemical Engineers72, 182-193.

  • Section 7.4. please explain more each section and compare recent studies together.
  • Section 9. It is supposed to add “future prospects to this section”.
  • I recommend replacing too old references with newly published references.

Reviewer 2 Report

The manuscript is well-organized, and it contains interesting view point to the readers of the journal.

However, they include non-soluble factors, such as receptors, into their "soluble factors" title, which should be changed.
